# Exploring the clinical context of adopting an instrumented insole: a qualitative study of clinicians' preferences in England

Denise Lin, Enrica Papi,[iD] Alison H McGregor

Department of Surgery and Cancer, Imperial College London, London, UK

**Correspondence to**
Dr Enrica Papi;
e.papi@imperial.ac.uk

## ABSTRACT

**Objectives** This study explores clinicians' views of the clinical uptake of a smart pressure-sensing insole, named Flexifoot, to enhance the care and management of patients with osteoarthritis (OA). Clinicians are key users of wearable technologies, and can provide appropriate feedback for a specific device for successful clinical implementation.

**Design** Qualitative study with in-depth, semi-structured interviews, analysed using inductive analysis to generate key themes.

**Setting** Conducted in a University setting.

**Participants** 30 clinicians were interviewed (11 physiotherapists, 11 orthopaedic surgeons, 5 general practitioners, 3 podiatrists).

**Results** All clinicians regarded Flexifoot to be useful for the care and management of patients in adjunction to current methods. Responses revealed four main themes: use, data presentation, barriers to use and future development. Flexifoot data were recognised as capable of enhancing information exchange between clinicians and patients, and also between clinicians themselves. Participants supported the use of feedback for rehabilitation, screening and evaluation of treatment progress/success purposes. Flexifoot use by patients was encouraged as a self-management tool that may motivate them by setting attainment goals. The data interface should be secure, concise and visually appealing. The measured parameters of Flexifoot, its duration of wear and frequency of data output would all depend on the rationale for its use. The clinicians and patients must collaborate to optimise the use of Flexifoot for long-term monitoring of disease for patient care in clinical practice. Many identified potential other uses for Flexifoot.

**Conclusions** Clinicians thought that Flexifoot may complement and improve current methods of long-term patient management for OA or other conditions in clinical settings. Flexifoot was recognised to be useful for objective measures and should be tailored carefully for each person and condition to maximise compliance. Adopting the device, and other similar technologies, requires reducing the main barriers to use (time, cost, patient compliance) before its successful implementation.

### Strengths and limitations of this study

► This was the first qualitative study to specifically explore clinicians' views on implementing an in-house smart, flexible, pressure-sensing insole tool into clinical practice for patients with osteoarthritis.
► The views of clinicians were fully explored with in-depth interviews, and analysed with inductive thematic analysis, giving rise to detailed suggestions to optimise the device's role alongside current strategies in patient care.
► Clinicians were unable to use the device themselves prior to the interviews and responses were based on one single demonstration and explanation of the tool.
► Clinicians had a varied level of experience and familiarity of wearable technologies between them, influencing their perspectives.

## INTRODUCTION

Osteoarthritis (OA) is one of the most common long-term musculoskeletal diseases, and cause of pain and functional disability.[1] Individuals who have sustained a knee injury, such as anterior cruciate ligament injuries, are three to six times more likely to develop knee OA.[2–4] In these patients, diagnosis occurs ~10 years prior to those without a previous injury,[3 4] calling for long-term management of such conditions. Current clinical guidelines recommend physical activity to delay surgical intervention, that is, known to have a limited lifespan and in many instances poor reported patient outcomes,[5–8] despite the belief that joint replacement is one of the most successful surgical procedures. Conversely, poor patient compliance limits long-term exercise benefits for OA, and many disregard the benefits of exercise.

Pain and gait changes are reasons why OA patients primarily visit clinicians. Gait analysis helps to establish OA diagnoses, severity and biomechanics underpinning musculoskeletal disorders.[9] In clinical settings, however,

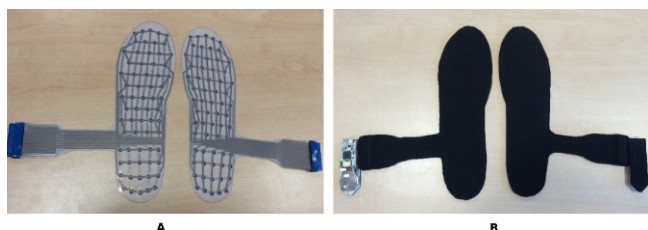

**Figure 1** (A) Layout of pressure sensors on the insoles with connectors for the circuit boards. (B) Insoles covered with neoprene with circuit boards for data transmission attached and ready to be inserted into shoes.

patient gait is observed by the clinician's eye, and self-reported questionnaires, such as the *36-Health Survey, Western Ontario and McMaster Universities Osteoarthritis Index, Oxford Knee Score* and *Knee Osteoarthritis Outcome Scores*, can assess OA severity.[9–11] Gait monitoring through clinician observation and patient questionnaires are prone to subjective responses, and therefore, are inadequate methods to quantify symptoms.

The emergence of wearable technologies can enhance current tools of physiotherapy, rehabilitation and daily monitoring of physical activity. Novel, portable wearable technologies offer a promising approach for use outside of the laboratory, to monitor functional changes in disease progression and activity levels. Nevertheless, the clinical implementation of wearable technologies is seemingly difficult. To enhance translation into clinical context, patient and clinicians' preferences have been explored in the past to determine the views and criteria of users for wearable technologies.[12] OA patients revealed that tracking disease progress was appealing and encouraged exercise.[13] Thirteen clinicians supported wearable technologies for OA patients clinically, but the preferences provided did not have specificity to one device.[14]

Within our group, we developed a smart, flexible, pressure-sensing insole, aptly named 'Flexifoot' (figure 1). Flexifoot generates plantar pressure readings from various foot regions. A high-resolution pressure map can be created from data that feed back wirelessly to a smartphone app, for the extrapolation of gait spatiotemporal parameters, centre of pressure and pressure distribution. Flexifoot, being portable and low cost to manufacture (~£20), in contrast to laboratory-based force platforms, allows for continuous data collection over a substantial number of gait cycles, for feedback to patients and clinicians as needed. Daily gait and pressure analysis can enable patients to monitor improvements and disease-related progression, as well as guide clinicians through treatment decisions. Flexifoot is yet to be validated; therefore, exploring users' preferences is beneficial for its ongoing development.

The ambiguity of previously obtained clinician preferences lacks the definitive feedback required to improve the design of a specific tool. The lack of specificity can be addressed by probing more into the details of the clinical implementation of Flexifoot for OA and other disorders.

The diagnosis and management of OA involves a multitude of healthcare professional types and it is therefore important to understand how Flexifoot could best address their requirements to inspire design and outcome measures which will facilitate clinical uptake. The aims of this study were to explore the clinicians' preferences for their use of Flexifoot and to identify specific parameters to be measured by the tool to foster improvements, and ultimately enhance OA patient care.

## METHODS
### Study design
The study was a qualitative study based on in-depth semi-structured interviews with 30 clinicians.

### Participants
Thirty clinicians (18 males and 12 females, aged 21–57 years), including 11 physiotherapists, 11 orthopaedic surgeons, 5 general practitioners (GPs) and 3 podiatrists, were recruited for one-to-one interviews.

Clinicians who had previously or currently worked for the Imperial College Healthcare National Health Service (NHS) Trust were invited via telephone and email invitations to partake in our study. The recruitment began from October 2015 and interviews occurred until September 2017, when data saturation occurred.

At the time of clinician interviewing, the healthcare professionals practiced among private and NHS settings within London and Greater London, one in Hereford, one in Cheltenham and one in Liverpool. They had from 4 months up to 28 years of experience within current specialities.

### Interviews and data
The interviews were performed by researchers (DL and MG) in person, except for 2, which were conducted over the telephone due to scheduling constraints. The researchers did not have any personal relationships with the study participants, and the group had prior experience in conducting qualitative studies.[13 14] Face-to-face interviews were audio recorded and transcribed afterwards.

Prior to each interview, participants' consent was obtained and researchers explained project aims, described Flexifoot and showed a prototype to each clinician (except in telephone interviews). Open-ended questions prompted clinicians to explore perspectives regarding the relevance of Flexifoot in clinical practice. The interview questions (online supplementary file 1) highlighted Flexifoot's clinical influence, specific measurements, data presentation preferences and gave scope for feedback and improvement.

The interview verbatim transcriptions were analysed using inductive thematic analysis,[15] without prior theoretical influences, whereby key findings were analysed and collated into early themes by DL and EP separately. DL and EP then checked each other's data and themes,

ensuring consistency and the generation of recurrent key themes.

## Patient and public involvement

While patients and users were not directly involved in the design of this study, this study arose from previous work where patients highlighted that their views were not considered in the design of novel wearable devices, thereby limiting uptake and translation.[13 14] This study directly focuses on care practitioners' preferences and requirements.

## RESULTS

The semi-structured interviews opened with questions to determine healthcare backgrounds of clinicians and the relevance of wearable technologies within their profession. Twenty-four out of 30 clinicians were aware of wearable technologies, and 4 used them for patients.

Inductive analysis[15] of interview responses revealed four main themes, with subthemes: use (applications, specific measurements, duration of wear), data presentation (data access, visual presentation, frequency of data), barriers to use and future development. The former three themes surfaced from specific interview questions asked. The latter was brought about after clinicians offered feedback as to how Flexifoot could be improved. The themes will be hereinafter described and verbatim quotes are indicated by: PT (physiotherapists), OS (orthopaedic surgeons), P (podiatrists) and GPs, followed by randomly assigned numbers.

### Uses

#### Applications

The main uses of Flexifoot identified by clinicians are shown in box 1.

All groups of clinicians recognised Flexifoot as an objective outcome measure tool to monitor various parameters. Twenty-three clinicians expressed that Flexifoot objective measures are useful to assess symptoms, and progress before and after medical intervention.

> It would be useful as an objective outcome measure for change…assess patients at time intervals for pre- and post-surgery. PT7

> This would be very interesting for research or pre- and post-surgery because you'd be able to monitor

---

**Box 1    The main five applications of Flexifoot in clinical practice identified by 30 clinicians**

Main applications of flexifoot in clinical practice:
► Assessing efficacy of treatment (pretreatment and posttreatment).
► Monitoring disease progression.
► Feedback for patients and other clinicians.
► Monitoring activity levels and compliance.
► Screening test to support future management.

---

and look at change…it would definitely be useful as a follow-up guidance to surgical correction. OS11

Twenty-one clinicians felt that objective data can reinforce clinical interpretations and enhance information exchange between healthcare professionals. Moreover, real-time objective feedback can help to visually demonstrate the problem and solutions to patients.

> For us feeding back to the surgeons…you can be a bit more accurate about what it is that you're saying. PT11

> It might be useful to demonstrate to the patient what some of their symptoms are. To give them a visual representation of that, I could show them this while they're walking. OS10

> It's important for the patients to visualise what the problem is…as a relatively low-grade, without major intervention, you could do a lot with it to see how to correct problems objectively…It may then allow them to see visually what the issue is, so that if they correct it with the help of someone. GP2

Clinicians recognised Flexifoot as a self-management tool: rehabilitation targets can be set by patients themselves, or by clinicians, and motivate patients.

> For patients to use at home as a rehabilitation tool to set targets or goals they can monitor themselves. PT8

> Anything that can give feedback to the patient themselves, to become more active and more healthy, then that could be of benefit, not necessarily to me, but to the patient. OS9

Seventy-five per cent of the clinicians supported Flexifoot measuring compliance to clinical advice.

> It would be really useful in terms of keeping a diary of what your patients are doing, especially with the osteoarthritis patients. PT5

> It is helpful if you have any doubts as to whether the patients are being compliant, if they are doing too much or too little activity. OS10

Clinicians can use the feedback as a screening tool, and to help determine the next steps for patient management.

> You could use that as some kind of screening test…do they really need to have a knee replacement yet? OS1

> In conjunction with physiotherapists…so if you were trying to get them to do a particular rehabilitation programme. Monitor what they're doing, that might be very useful. OS6.

However, GPs felt that Flexifoot was a tool to be used more by patients, rather than by GPs for planning patient care: '*an intervention that's positive for the patient, as opposed to this being an investigation*' *GP2,* and that feedback would be better interpreted by clinicians with greater musculoskeletal knowledge.

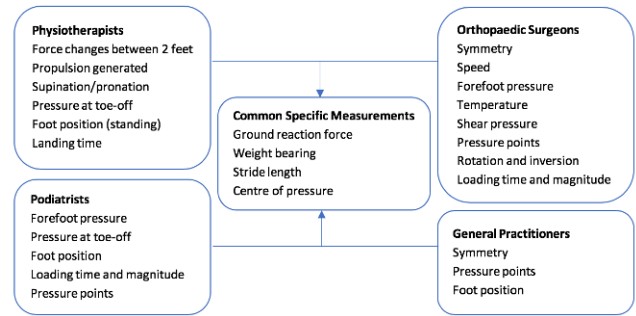

**Figure 2** Clinician preferences for the specific measurements that they would like Flexifoot to obtain.

I can't see any acute use for it that's going to change the patient's management right this second now. GP2

I'm not sure whether this would change my management for the conditions. GP5

Moreover, various participants in all clinician groups agreed that Flexifoot may be more effective not as a standalone device, but to enhance current methods for diagnosis and prognosis, as an '*adjunctive thing to what I already have*' P3.

### Specific measurements

Due to the vast array of parameters that can be analysed after Flexifoot use, it was important to determine the most clinically appropriate specific measurements. Participants were prompted by suggestions of symmetry, stride length, centre of pressure, pressure profiles or ground reaction force. The ideal specific measurements differed slightly between clinician groups due to varying levels of knowledge, but there was a convergence of agreed parameters to be measured by Flexifoot in all clinician groups (figure 2). GPs provided fewer preferable specific measurements due to gaps in their specialist knowledge regarding musculoskeletal disorders.

The clinicians who supported the monitoring of activity levels (type and number of steps) demonstrated its importance for non-OA conditions too.

It would give you an idea of their daily routine and exercise levels and things, particularly if they've got other conditions such as diabetes or cardiovascular disease which can be improved by exercise. GP3

Two clinicians exhibited a cautious view regarding tracking patients' activity outside of clinical environments:

You've got to be careful with monitoring… you don't want engender this sort of 'we're watching you' big brother idea. PT3

I do think that there could be some, a little degree of patients being suspicious of you checking up on them and they may question the clinician—why can't you trust what you're telling them, why do they need to see what I'm doing? GP2.

### Duration of wear

The duration of Flexifoot wear suggested by clinicians varied from single usage at clinic appointments, up to long-term periods of over 1 year, implying a range of uses. The nature of the tool being able to offer real-time advice allows for its acute use in the clinic. The prolonged use of Flexifoot may be appropriate for patients with chronic diseases, or postoperatively in those undergoing surgical interventions. The disparity between preferred lengths depended on the rationale for patient use. Seven clinicians revealed that the duration of Flexifoot wear would be dependent on what outcomes were to be achieved (PT1, PT6, PT8, P2, OS1, OS6), type of injury (PT10), and patient age and compliance (GP5).

If we were using it as a 'how do they move', we want a picture of their footprint, 5 min as they walk along the corridor. PT1

During periods of activity, if they're experiencing pain or there is a particular challenging part of their day…short bursts of time which are critical to look at, so definitely not all day. OS3

However, long-term monitoring was preferred by most clinicians, where data collection would span over days, weeks, months and years, or '*for as long as it took to establish a meaningful change*' PT3.

For 24 hours or a few days…you'd want the results of this to reflect what they normally do. GP3

A short period of time using the device but at longer intervals between uses was also suggested:

Snapshots at certain periods, much like we do at follow-up, at 6 weeks, 3 months, and a year. OS4

To maximise the personalised use of Flexifoot, clinicians and patients must collaborate to choose an appropriate duration of wear for each particular case.

### Data presentation
#### Data access

Fifty per cent of the clinicians would prefer to access the data by logging into a system, and some expressed that it should be integrated among patients' current notes for information crossover.

If there was a way of tying in the results so that when you log in and click on the patient, it came up on their results. That would be the most ideal way. GP3

Five clinicians preferred the data via email, although three clinicians stated that maintaining patient confidentiality was paramount.

Emailed is easier, but then log-in would be better because safety aspects and confidentiality. P2

#### Visual presentation

The data should be concise so that clinicians can quickly interpret data. The clinicians expressed differing opinions

after they were prompted with suggestions of graphs, tables and pictures. Therefore, numerous display options should be available to accommodate for all preferences.

> We need to have a summary that is brief, because you don't want to look through tons and tons of data. Then, if you wanted to look into more detail, then there should be the option. GP4

The data should be easily comprehended by patients too. A visually appealing approach with an additional colour-system scheme can enhance patient understanding.

> Patients want to know as well, they want a variation that's patient friendly…an easier format for patients to understand. OS6.

The addition of normal reference ranges alongside objective measures allows for the comparison of patient parameters versus reference data; this will enhance user-experiences and allow for goals to be set.

> Patient specific—it would be graphs. It would be nice to have a normal distribution and see where they fit inside the normal distribution. OS9

### Frequency

The clinicians want to receive and access data at appropriate times, such as immediately before or during appointments. The majority (28/30) of clinicians were unlikely to monitor patients outside of the clinic due to time restraints.

> I'd access the data just prior to the patient coming in or during that appointment. We would only really have time to monitor or review the data when the patient is actually here. PT5

> Real-time is helpful for the patient in a therapy session. If they had information once a week on how they're doing. Alerts are ideal, but there is no time to check it. PT10

The generation of data should occur in a timely fashion, *'correlated with the patient's clinic appointment' OS9.* The frequency of data received should depend on personal clinician preferences and the purpose of Flexifoot use.

> There should be an option for the data to be accessible when you want it and you choose. GP5

### Barriers to use

Clinicians highlighted obstacles to implementing Flexifoot (box 2).

Fifty per cent of the clinicians illustrated that time restraints were the largest concern of Flexifoot use clinically, including time taken for initial patient assessments, device introduction and data generation.

> For me to use this for one patient, explain how to use it and monitor their activity is probably unlikely and

---

**Box 2   The barriers to the use of implementing Flexifoot in clinical practice identified by 30 clinicians**

Barriers to the use of Flexifoot:Barriers to the use of Flexifoot:
- ► Time.
- ► Cost and availability.
- ► Influence on practice.
- ► Training/education required.
- ► Patient compliance.
- ► Hygiene control.

---

unrealistic given the general practice workload and increasing demands on GPs. GP2

> That is the difficulty with this, it is an additional investigation that we need to spend time assessing. OS11

The second most identified barrier was cost— *'it has to be suitably priced…that an average practice can afford' P1.*

The inability of clinicians and patients to interpret data and the training required was another issue. Five GPs stressed that Flexifoot was too specialised for their environment.

> Patients might not understand the data…people are not familiar with technologies, but this will be less of a problem in the future. PT9

> If it's mainly biometrics and gait analysis…I don't think that I would see this as being within a GP's remit so much. The biggest barrier for me not using it is, identifying how the information it gives would fall into my remit, and how it would influence my practice. GP3

Poor patient compliance also hinders Flexifoot's prospective use.

> Patients are so unreliable and I wouldn't be confident that they would remember to transfer it to another pair of shoes or if they take their shoes off and we'd not be tracking anything. OS3

Six clinicians identified hygiene and infection risks if the device was used for long periods, or between different patients. Flexifoot use between multiple patients could be more economically practical, however.

> I suppose you've got questions of hygiene…you'd need a material you can wipe and maybe some way of cleaning them really well, so they would be to an infection control standard. If you can use them more than once, it would be cost effective. P3

### Future development

Following the responses to the set questions, clinicians offered suggestions on how to optimise Flexifoot for successful clinical uptake.

Parameters that would be ideally measured using Flexifoot (figure 2) could be adapted to measure more factors and expand patient target audiences. Clinicians

suggested Flexifoot use in Parkinson's disease, peripheral neuropathy in diabetes, chronic pain conditions, obesity, hemiplegia and tendinopathies.

> Pressure profiles are good, but it'd be really great to measure shear and temperatures in the foot… it would be better and more useful for people with plantar foot pain. Diabetics—it would be great. OS11

> Design changes were also proposed, '*maybe this ribbon (ankle strap) could be a bit smaller because it might irritate someone on the side and it might artificially affect their gait' OS10.*

## DISCUSSION

A technology must be user-friendly to optimise its efficacy and sustainability.[16 17] We conducted structured interviews with clinicians to guide the development of our novel technology, Flexifoot, towards clinical uptake. The clinicians expressed numerous advantages of adopting Flexifoot into healthcare settings and barriers, indicating strategies for future improvement. The main advantage recognised by clinicians was the ability of Flexifoot to generate quantitative data, that can be used for monitoring and feedback in various clinical contexts.

Clinicians implied that Flexifoot would not replace current clinical tools, but instead complement them. The ambulatory quantitative data can support existing OA diagnostic and management tools, and help to improve the reliability of clinical decisions. The ambulatory monitoring of disease progression, alongside patients' responses to treatment, has been considered useful.[18] Other rehabilitation technologies that motivate and offer objective feedback have been associated with long-term benefits and good physical fitness levels.[19] Despite this, some lower limb wearable technologies for rehabilitation have been described to have limited efficacy for the improvement of activity levels, but mainly due to poor research methodologies used in the past.[20] However, there is still a demand for self-management of rehabilitation with feedback using shoe insole pressure sensors.[21] The contradictory results in the literature regarding wearable technologies calls for a study that explores the users' perspectives to maximise acceptance.

Moreover, objective feedback can enable a more efficient exchange and handover of patient information between clinicians.[22] A clinician may use the tool as a screening approach in adjunction to current methods, for more reliable results and subsequently refer the patient onto a specialist.[14] Also, the clinicians recognised that the data can reinforce their dialogue with patients, making patients more aware of their problem. Awareness and feedback was seen as a way to enhance patient self-management. Tracking activity levels and feedback engages patient involvement in their own care, and is useful for other non-musculoskeletal chronic conditions too.[23 24]

Clinicians, in our study, indicated how Flexifoot could be a feedback tool for patients. The data being available to patients allows for greater independence in self-monitoring and feedback of their own diseases in familiar environments outside of clinics.[18] Home-based training and monitoring devices showed higher patient satisfaction compared with similar care within clinics.[25] Self-management can also reduce economic burdens as the technology can educate OA patients, improve outcomes and reduce hospital visits.[18 26]

The uses and specific measurements suggested by clinicians was greatly dependent on the type of clinician and specific patient cohorts, which is also the case for other musculoskeletal interventions.[27] The GPs felt that they had insufficient gait analysis knowledge, and that the device was presently too specialist for general practice. GP environments may be inappropriate since it comprises of too broad a range of patients. Instead, they indicated that Flexifoot would be better suited for clinicians who follow-up patients more regularly. This was reiterated by physiotherapists, orthopaedic surgeons and podiatrists, who are better equipped in musculoskeletal fields and expressed positivity for Flexifoot's clinical uptake.

All clinicians indicated that the system and data output should be easy to use and interpret. Ease of use of wearable sensors for clinicians was reported in the past to facilitate their adoption into clinical settings.[28] The prospect of a log-in software system for Flexifoot data was more popular than receiving results via email. The interface should be integrated alongside current patient records for information crossover and a choice of data presentation styles should be available. The material should be presented alongside normal reference ranges, for comparisons and targets to be made. Data accessibility and presentation should be understood by patients too—this is key for patient acceptance and accessibility.[29] Clinicians' views obtained in another study recognised that shorter, simpler and more concise data as educational material are preferable for patients, but that detailed data should be fully available too.[30] This agrees with our findings: participants expressed the possibility of having access to more detailed data if needed, while avoiding scanning excessive data beyond their understanding. Full data measurements could be stored, however, for more skilled users in research settings.[31]

Furthermore, shortcomings of Flexifoot were recognised by the clinicians which may explain why the practical application of similar insole monitoring devices have not been successful in the past.[32]

The clinicians' continuous workloads means that using real-time data from Flexifoot is only feasible prior to or during appointments in the presence of patients. The real-time data and automatic alerts may be more useful for patient users for receiving feedback, which continues to motivate them.[13] Clinicians are reluctant for the introduction of new tools because they can disrupt time-pressured practice schedules, and the time required to train them to use Flexifoot must also be considered.[33] In the past, numerous medical wearable technologies for a range of users have failed to meet the criteria of being simple and

powerful in terms of data output and energy consumption.[32] However, although clinicians may perceive new tools as a hindrance, a study showed that adults suffering from OA felt that more novel approaches could be implemented for the management of their condition.[34 35] The clinical efficacy of Flexifoot must be therefore fully established, such as through patient usability testing, before clinicians can adopt it as a method worthy of appointment time.

High costs also limit new technology implementation within health services. The current expense to manufacture one Flexifoot device is low, but one tool per patient may be economically impractical. The recycling and reuse of devices between patients may reduce costs, but increases hygiene and infection risks, expressed by six clinicians. Introducing hygienic procedures prior to and after Flexifoot use could enhance its reusability and cost-effectiveness. Thus, the cost-effectiveness, with reference to current treatment guidelines, should be further investigated. Patient compliance is also an issue since long-term wearable devices require adequate patient acceptability. However, from a similar study we conducted that explored patients' views, all participants were keen for the uptake of wearable technologies.[13] Moreover, effective clinician–patient communication can determine patient cohorts supportive of self-management, and those more likely to adhere to using Flexifoot.[13 14 36] Clinicians can promote the relevance of sensor technologies for patients' care, and hence boost compliance.[12 37] The identified issues surrounding Flexifoot are apparent in other wearable technologies too.[27] The replication of problems between devices implies a necessity for new approaches in encouraging patients' compliance and appeal for novel strategies. The findings that emerged from this study can be translated to other similar technologies to promote their clinical uptake and foster new developments.

The study limitations involved the clinicians' varied levels of experience and familiarity of instrumented insoles, and that 27 out of 30 interviewees were based in the London area. The clinicians had not used Flexifoot, and telephone interviews could not view the device, but detailed descriptions and commercially similar devices that could be found online were provided prior to interviews. Future studies would involve clinicians' use of Flexifoot beforehand. Moreover, clinicians who were well-informed around the subject were perhaps biased to initially participate and express positivity. However, the interviews were confidential and honest feedback was encouraged.

## CONCLUSIONS

In conclusion, the clinicians considered Flexifoot to be a useful tool that could be used in adjunction to current approaches, in a long-term, follow-up setting to support and improve patient care. The clinicians' preferences exhibited numerous ways in which Flexifoot can be useful for patients with OA or other conditions. The measured parameters should be selected according to patient-specific cases, and delivered in a concise manner through a secure interface. A choice of data outputs should be offered to cater for all users. The challenges of time, cost, infection control should be addressed, alongside the clinical efficacy and cost-effectiveness for the clinical adoption of Flexifoot and similar technologies.

**Acknowledgements** We would also like to thank MSc student Mary Goodwin for her contribution with data collection.

**Contributors** DL, EP and AHM conceived and designed the study. DL carried out the interviews. DL and EP analysed and interpreted the data. DL and EP drafted the manuscript. All authors read, edited and approved the final version of the manuscript. All authors had full access to all of the data in the study and can take responsibility for the integrity of the data and the accuracy of the data analysis.

**Funding** EPSRC and Wellcome Trust.

**Competing interests** None declared.

**Patient consent for publication** Obtained.

**Ethics approval** The study was reviewed and approved by Imperial College London Ethics Research Committee.

**Provenance and peer review** Not commissioned; externally peer reviewed.

**Data sharing statement** No additional data are available.

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
