## [Reviewer comments · BMJ Open]

ARTICLE DETAILS

TITLE (PROVISIONAL)	Exploring the clinical context of adopting an instrumented insole: a qualitative study of clinicians' preferences in England
AUTHORS	Lin, Denise; Papi, Enrica; McGregor, Alison

VERSION 1 - REVIEW

REVIEWER	Dr Jack Parker School of Health and Related Research, University of Sheffield England
REVIEW RETURNED	28-Jun-2018

GENERAL COMMENTS	This is an interesting and much needed area to be explored. The authors present an insight into the clinical perspectives of wearable technology for OA from differing clinical professionals which provides a well rounded view of the positives and negatives of the clinical utilisation of the Flexifoot. However, I recommend some alterations that might enhance the paper: The literature review is somewhat limited. This section could reflect on the broader issue of wearables such as other devices and the efficacy of these devices (see Powell et al, J Med Internet Res 2016;18(10): e259) and the efficacy of insole pressure sensors (see Munoz-Organero et al, Sensors. MDPI. Volume 16. October 2016. Sensors 2016, 16(10), 1631. The method of analysis could be more clearly described. For example, a description of the process of analysing data such as whether the themes were a priori or if a more inductive approach was adopted. How did the themes emerge? How were they validated and by whom? Was there any respondent validation? Limitations: For telephone interviews it is unclear how participants were able to comment on the Flexifoot. Also there is no reflexivity account. Overall, an interesting and insightful paper
--

REVIEWER	Bert Aertgeerts KU Leuven, Belgium
REVIEW RETURNED	16-Sep-2018

GENERAL COMMENTS	As said in the attached file, I could not have enough insights in the methodological issues to perform a soundful critical appraisal of your paper Methods-COREQ
--

	Domain 1: Research team and reflexivity	
	Personal characteristics	
	Interviewer/facilitator	check
	Credentials	unclear
	Occupation at the time of the study	unclear
	Gender	check
	Expertise and training	unclear
	Relationship with participants	
	Relation established	unclear
	Participant knowledge of the interviewer	unclear
	Interviewer characteristics	unclear
	Domain 2: study design	
	Theoretical framework	
	Methodological orientation and theory	deductive
	thematic analysis	
	Sampling	unclear
	Method of approach	mail, telephone
	Sample size	30 clinicians
	Non-participation	unclear
	Setting	
	Setting of data collection (greater) London	HCP within NHS
	Presence of non-participants	unclear
	Description of sample	check
	Data collection	
	Interview guide	unclear
	Repeat interviews	unclear
	Audio/visual recording	check
	Field notes	unclear
	Duration	unclear
	Data saturation	unclear
	Transcripts returned	unclear
	Domain 3: Analysis and findings	
	Data analysis	
	Number of data coders	3 (DL, MG, EP)
	Description of the coding tree	No
	Derivation of themes	unclear
	Software	No
	Participant checking	No
	Reporting	
	Quotation presented	Yes/Yes
	Data and findings consistent	Yes
	Clarity of major themes	Yes
	Clarity of minor themes	No

	Besides the relevance of this research, the methodology of this qualitative paper is very poor using the COREQ as a critical appraisal instrument.
--	--

REVIEWER	Leanne Hassett The University of Sydney
REVIEW RETURNED	18-Nov-2018

GENERAL COMMENTS	This is an interesting paper, however it requires further detail for the reader to understand the relevance of the findings & the rigor of the study. Specifically:  1) abstract: themes should be included in results section. 2) More information is required about the Flexifoot device. Is there a published paper to refer to? If not, more information is required re: cost and where it is up to in development. A picture could be helpful. The aim of the paper seems to need refinement to include that you are exploring clinician views to aid in development of Flexifoot. 3) It would be good to add a sentence to explain deductive analysis for the reader. From my reading it says typically you start with a theory and use the data to confirm or refute the theory. I cannot see you have done this in your analysis. Also it is unusual to me that you include quantitative data in your results e.g. "23 clinicians....." 4) There is information missing from your paper describing your study that helps determine the rigor. Suggest using the COREQ checklist to ensure all methods are described appropriately e.g. screening/recruitment/sampling of participants; interview length; relationship with interviewers and interviewees; member checking. 5) Discussion section needs clearer information about what is needed for this device to be used in practice such as patient usability testing. You mention compliance may be an issue but patient acceptability seems key to successful implementation, particularly if long wear-time is wanted.
--

VERSION 1 – AUTHOR RESPONSE

Reviewer(s)' Comments to Author:

Reviewer: 1

Reviewer Name: Dr Jack Parker

Institution and Country: School of Health and Related Research, University of Sheffield, England

Please state any competing interests or state 'None declared': None declared

Please leave your comments for the authors below

This is an interesting and much needed area to be explored. The authors present an insight into the clinical perspectives of wearable technology for OA from differing clinical professionals which provides a well rounded view of the positives and negatives of the clinical utilisation of the Flexifoot. However, I recommend some alterations that might enhance the paper:

The literature review is somewhat limited. This section could reflect on the broader issue of wearables such as other devices and the efficacy of these devices (see Powell et al, J Med Internet Res 2016;18(10): e259) and the efficacy of insole pressure sensors (see Munoz-Organero et al, Sensors. MDPI. Volume 16. October 2016. Sensors 2016, 16(10), 1631.

The method of analysis could be more clearly described. For example, a description of the process of analysing data such a whether the themes were a priori or if a more inductive approach was adopted. How did the themes emerge? How were they validated and by whom? Was there any respondent validation?

Limitations: For telephone interviews it is unclear how participants were able to comment on the Flexifoot. Also there is no reflexivity account.

Overall, an interesting and insightful paper

We would like to thank you for reviewing and giving these suggestions.

- We have modified the literature review (Paragraph 1-2 in Discussion section) to include these studies and have considered that the contradictory results within various studies exploring the efficacy of wearable technologies highlight the fact that our study is useful to explore a users' perspective to increase the acceptance of such devices.

- Furthermore, we have more clearly described our methodology and analysis. We have adjusted the terminology and it is in fact inductive analysis that was conducted with the extraction of main themes without the prior theoretical influences. Researchers, previously experienced in conducting qualitative studies, separately analysed data and then confirmed each other's results too to ensure consistency between themes and results. This has been included in the Methods section.

- A reflexivity account has been added within the Methods section. The Limitations now highlight that in the telephone interviews, prior detailed descriptions of the Flexifoot were given and its function was fully described. We also referred to other similar commercially available devices for the respondents to look up online if needed.

Reviewer: 2

Reviewer Name: Bert Aertgeerts

Institution and Country: KU Leuven, Belgium

Please state any competing interests or state 'None declared': None declared

Please leave your comments for the authors below

As said in the attached file, I could not have enough insights in the methodological issues to perform a soundful critical appraisal of your paper. Please see attached file (BMJ Open review.pdf)

The SRQR checklist has now been provided alongside this study and the methods section has been modified, indicating appropriate rigor of the methodology.

Reviewer: 3

Reviewer Name: Leanne Hassett

Institution and Country: The University of Sydney

Please state any competing interests or state 'None declared': None declared

Please leave your comments for the authors below

This is an interesting paper, however it requires further detail for the reader to understand the relevance of the findings & the rigor of the study. Specifically:

- 1) abstract: themes should be included in results section.
- 2) More information is required about the Flexifoot device. Is there a published paper to refer to? If not, more information is required re: cost and where it is up to in development. A picture could be helpful. The aim of the paper seems to need refinement to include that you are exploring clinician views to aid in development of Flexifoot.
- 3) It would be good to add a sentence to explain deductive analysis for the reader. From my reading it says typically you start with a theory and use the data to confirm or refute the theory. I cannot see you have done this in your analysis. Also it is unusual to me that you include quantitative data in your results e.g. "23 clinicians....."
- 4) There is information missing from your paper describing your study that helps determine the rigor. Suggest using the COREQ checklist to ensure all methods are described appropriately e.g. screening/recruitment/sampling of participants; interview length; relationship with interviewers and interviewees; member checking.
- 5) Discussion section needs clearer information about what is needed for this device to be used in practice such as patient usability testing. You mention compliance may be an issue but patient acceptability seems key to successful implementation, particularly if long wear time is wanted.

Thank you for your suggestions, and we have now:

- 1) Included the themes of the results within the abstract
- 2) Provided more information about Flexifoot. An image has been added (Figure 1) and its price point has also been included. We have also noted the devices' current stage of development and that we will ultimately use the clinicians' preferences to also improve the development of Flexifoot. This can be found in paragraph 4 of the Introduction.
- 3) The type of analysis has been adjusted to inductive analysis so it has been now appropriately defined within the methods section. Inductive analysis involved the analysis whereby we extracted key themes without any prior theoretical influences.
- 4) We have added in more additional information regarding the researchers and reflexivity accounts. Also, the SRQR checklist has now been provided alongside this study which highlights appropriate rigor of the methodology.
- 5) The discussion section has been modified, specifically within paragraph 4 of the Discussion section. Here, we highlight that patient usability testing is a good method to establish clinical efficacy of Flexifoot. We also further considered the issues of patient compliance that you mentioned, and have related this back to a similar qualitative study that our group conducted which reiterated that patients supported the use of wearable technologies for osteoarthritis.

VERSION 2 – REVIEW

REVIEWER	Leanne Hassett The University of Sydney
REVIEW RETURNED	29-Dec-2018

GENERAL COMMENTS	The manuscript is much improved with the new edits. A few minor points to consider: 1) I think the conclusions could be strengthened by adding in some more generalizable points in regards to use of insole monitoring devices given the Flexifoot is not commercially available, thus there needs to be research/clinical/development implications for the reader to consider.2) Page 20, lines 57-58: the new sentence. Suggest add: “.....preferences is beneficial for its ONGOING development.”3) Page 22, section “Participants”: state sampling technique. Sounds like convenience sampling. More detail needed about sampling. Did you recruit from ?number of hospital or health districts? It is still not clear how and where you found your sample other than by telephone and email. Did you email all GP practices in a particular county/health district etc?4) Tables 1 and 2: Don’t need 2nd row, it just repeats information in row 1. I also question need for tables at all, the information could easily be described in a sentence.5) Discussion: Paragraph 2, very long. The added sentences about stroke seem a little out of place. Page 27, line 53, “The uses and specific measurements.....” this is a new section; the paragraph could perhaps be split at this point.
---

VERSION 2 – AUTHOR RESPONSE

Reviewer(s)' Comments to Author:

Reviewer: 3

Reviewer Name: Leanne Hassett

Institution and Country: The University of Sydney

Please state any competing interests or state 'None declared': None declared

Please leave your comments for the authors below

The manuscript is much improved with the new edits. A few minor points to consider:

1) I think the conclusions could be strengthened by adding in some more generalizable points in regards to use of insole monitoring devices given the Flexifoot is not commercially available, thus there needs to be research/clinical/development implications for the reader to consider.

The shortcomings of Flexifoot, and other similar wearable technologies such as insole monitoring devices, have been expanded upon now in the discussion. General points have now been added to explain in greater detail the overall problems that medical wearable technologies face: “In the past, numerous medical wearable technologies for a range of users have failed to meet the criteria of being simple and powerful in terms of data output and energy consumption. However, although clinicians

may perceive new tools as a hindrance, a study showed that adults suffering from osteoarthritis felt that more novel approaches could be implemented for the management of their condition.”

2) Page 20, lines 57-58: the new sentence. Suggest add: “.....preferences is beneficial for its ONGOING development.”

The word “ongoing” has now been added into this sentence.

3) Page 22, section “Participants”: state sampling technique. Sounds like convenience sampling. More detail needed about sampling. Did you recruit from ?number of hospital or health districts? It is still not clear how and where you found your sample other than by telephone and email. Did you email all GP practices in a particular county/health district etc?

Thanks for pointing this out. New comments have now been added into the “Participants” section which will hopefully provide some clarity to the sampling. Clinicians that had previously or currently worked for the Imperial College Healthcare National Health Service (NHS) Trust were invited via telephone and email invitations to partake in our study. We did not have a set number in mind but we continuously recruited and interviewed from October 2015, when data saturation occurred, then recruitment stopped.

4) Tables 1 and 2: Don’t need 2nd row, it just repeats information in row 1. I also question need for tables at all, the information could easily be described in a sentence.

Thanks for this comment. We feel as if the use of tables allows for the information to be summarised in a quick manner facilitating the reader with the understanding of the manuscript.

In both tables Row 1 is the caption of the table. This has now been removed from the actual table as this caused confusion and positioned before each table.

5) Discussion: Paragraph 2, very long. The added sentences about stroke seem a little out of place. Page 27, line 53, “The uses and specific measurements.....” this is a new section; the paragraph could perhaps be split at this point.

The sentences regarding the post-stroke lower limb rehabilitation have been edited and placed in a different place in the discussion for a more cohesive read, and the paragraphs here have also been broken down to allow for the discussion to flow better.

VERSION 3 - REVIEW

REVIEWER	Dr Leanne Hassett The University of Sydney, Australia.
REVIEW RETURNED	21-Feb-2019

GENERAL COMMENTS	All previous concerns have been addressed. One small typo in new sentence in discussion section: page 8, line 32: “.....there is a still a demand...” Remove "a" before "still".
--

VERSION 3 – AUTHOR RESPONSE

Reviewer(s)' Comments to Author:

Reviewer: 3

Reviewer Name: Dr Leanne Hassett

Institution and Country: The University of Sydney, Australia.

Please state any competing interests or state 'None declared': None declared.

Please leave your comments for the authors below All previous concerns have been addressed. One small typo in new sentence in discussion section: page 8, line 32: ".....there is a still a demand..."

Remove "a" before "still".

Thanks for spotting this typo. This has now been corrected as suggested